# Appraisal of Heavy Metals Accumulation, Physiological Response, and Human Health Risks of Five Crop Species Grown at Various Distances from Traffic Highway

Shakeel Ahmad [1], Fazal Hadi [1], Amin Ullah Jan [2], Raza Ullah [1,3], Bedur Faleh A. Albalawi [4] and Allah Ditta [5,6,*]

[1] Laboratory of Molecular Stress Physiology and Phytotechnology, Department of Biotechnology, Faculty of Biological Science, University of Malakand, Chakdara 18800, Pakistan
[2] Department of Biotechnology, Faculty of Science, Shaheed Benazir Bhutto University, Sheringal, Dir (upper) 18000, Pakistan
[3] Laboratory of Plant Molecular Biology and Biotechnology, Department of Biology, School of Arts and Science, University of North Carolina at Greensboro, Greensboro, NC 27412, USA
[4] Department of Biology, University of Tabuk, Tabuk 47512, Saudi Arabia
[5] Department of Environmental Sciences, Shaheed Benazir Bhutto University, Sheringal, Dir (upper) 18000, Pakistan
[6] School of Biological Sciences, The University of Western Australia, 35 Stirling Highway, Perth, WA 6009, Australia
* Correspondence: allah.ditta@sbbu.edu.pk or allah.ditta@uwa.edu.au

**Abstract:** Road surfaces and vehicular traffic contribute to heavy metals (HM) contamination of soil and plants, which poses various health risks to humans by entering the food chain. It is imperative to evaluate the status of contamination with HM and associated health risks in soils and plants, especially food crops. In this regard, five crop species, i.e., strawberry (*Fragaria ananassa*), wheat (*Triticum aestivum*), tomato (*Lycopersicon esculentum*), sugar cane (*Saccharum officinarum*), and tobacco (*Nicotiana tabacum*), were evaluated at 0–10, 10–50, and 50–100 m distance from the highway near the urban area (Takht Bhai) of Mardan, Khyber Pakhtunkhwa, Pakistan. Lead (Pb) and cadmium (Cd) accumulation, phenolics, carotenoids, chlorophyll, and proline contents in plant parts were assessed. Pb and Cd in plants decreased with an increase in distance. Pb was above the critical limit in all plants except wheat, Cd exceeded the permissible level of the World Health Organization in all plants except wheat and tomato. Pb and Cd were higher in strawberries. Tomato and strawberry fruits, tobacco leaves, and sugarcane stems showed higher Pb contents at a 0–10 m distance. Phenolic contents in leaves were higher than in roots. The target hazard quotient (THQ) in edible parts of most crops has been greater than one, which presents a threat to human health upon consumption. To the best of our knowledge, this study presents the first holistic approach to assess metal contamination in the selected area, its accumulation in field-grown edible crops, and associated health risk.

**Keywords:** lead; cadmium; crops; soil contamination; health risk index; phenolics; proline; urbanization

## 1. Introduction

Food security and safety have been a special concern worldwide due to the rise of natural lands contaminated with heavy metals (HM) and other classes of emerging contaminants, which are inextricably linked with human health [1–3]. The root causes of this issue are widely linked to the rapid pace of urbanization and land use for industrialization and roads, particularly in developing countries with high population growth [4]. Among various anthropogenic sources, road surfaces and vehicular traffic add HM and ultimately contaminate the soil [5,6]. Heavy metals such as lead (Pb), cadmium (Cd), copper (Cu), and Zinc (Zn) are released into the environment during various operations of road transport [7,8]. Road transport mainly deposits Pb and Cd from fuel burning, wear

out of tires, and leakage of oils, which contaminate soil and edible crops [9,10]. The bioaccumulation of these toxic metals in crop plants has received global attention because of their negative effects on human health [11–13] and phytotoxicity [14]. Pb in humans causes neurological disorders, anemia, hypertension, and impaired renal function [15]. Cd is mutagenic and carcinogenic and its elevated levels in human causes damage to the kidneys, liver, and bones [16]. In plants, HM induce certain effects such as growth reduction, nitrogen assimilation, the inhibition of chlorophyll, and enzyme activities [17–19]. HM (Pb and Cd) initiate oxidative stress and some plants combat the oxidative stress through anti-oxidative defense systems by producing phenolics, carotenoids [20–22], and free proline [23,24].

The edible crops grown in the agricultural fields near the roadside tend to accumulate HM and are considered a serious threat to human health [10]. Therefore, it is essential to assess the existing levels of accumulated metals in edible parts and their comparison to the safe levels/thresholds recommended by international organizations such as the World Health Organization (WHO). The permissible levels for Pb and Cd in edible plants are 0.3 mg kg$^{-1}$ and 0.1 mg kg$^{-1}$, respectively [25,26]. Numerous studies have been conducted on the perspective of metal contamination in edible plants and its adverse health effects on humans if consumed [27,28]; however, there is still a lack of insight to investigate the toxic metals in food commodities and their risk assessment in many developing countries [29]. In the same domain, qualitative and quantitative analysis of HM has not been focused and no efforts have been made to fully assess the associated human health risk [30]. As a proactive step to formulating suitable prevention strategies for soil contamination by HM, accurate mapping of pollutants in a given area is needed [8,10].

To the best of our knowledge, it is the first reported study to assess the concentration of toxic metal in roadside farmlands and edible crops and ascertain risk assessment in the selected study area, and this research is significant to lay the foundation for further studies regarding metals accumulation in edible plants growing in metal-contaminated sites near the traffic highways and other polluted areas. For this purpose, five important crops species, i.e., strawberry (*Fragaria ananassa*), tomato (*Lycopersicon esculentum* L.), wheat (*Triticum aestivum*), sugar cane (*Saccharum officinarum* L.) and tobacco (*Nicotiana tabacum*) growing under natural environmental conditions were selected and these crops are popularly cultivated in the study area under consideration because farmers prefer to grow them for their high income. Pb and Cd accumulation from HM-contaminated soils has been reported in strawberries [31], tomatoes [32], wheat [33], sugarcane [34], and tobacco [35]. The current study was conducted with an aim to (1) evaluate Pb and Cd concentrations in soil samples and in various parts of selected crop plants, (2) to investigate the antioxidants and biochemical parameters (phenolic, carotenoids, proline, and chlorophyll contents) in leaves and roots and to examine their correlation with metals concentration in plants (3) to compare Pb and Cd concentration in plants with WHO threshold level. (iv) to evaluate the risk assessment based on HRI and bio-concentration factor (BCF).

## 2. Materials and Methods

### 2.1. Site Description and Samples Collection

The selection of the study site was based on the distance from the highway in an urban area (Takht Bhai) of Mardan, Khyber Pakhtunkhwa, Pakistan (Figure 1). Mardan is located at 34.1989° N, 72.0231° E. The road is surfaced with tar coal and is experiencing a huge traffic density including trucks, passenger buses, private cars, and others. Three plots on the bases of distances (0–10, 10–50, and 100 m) from the main road were selected.

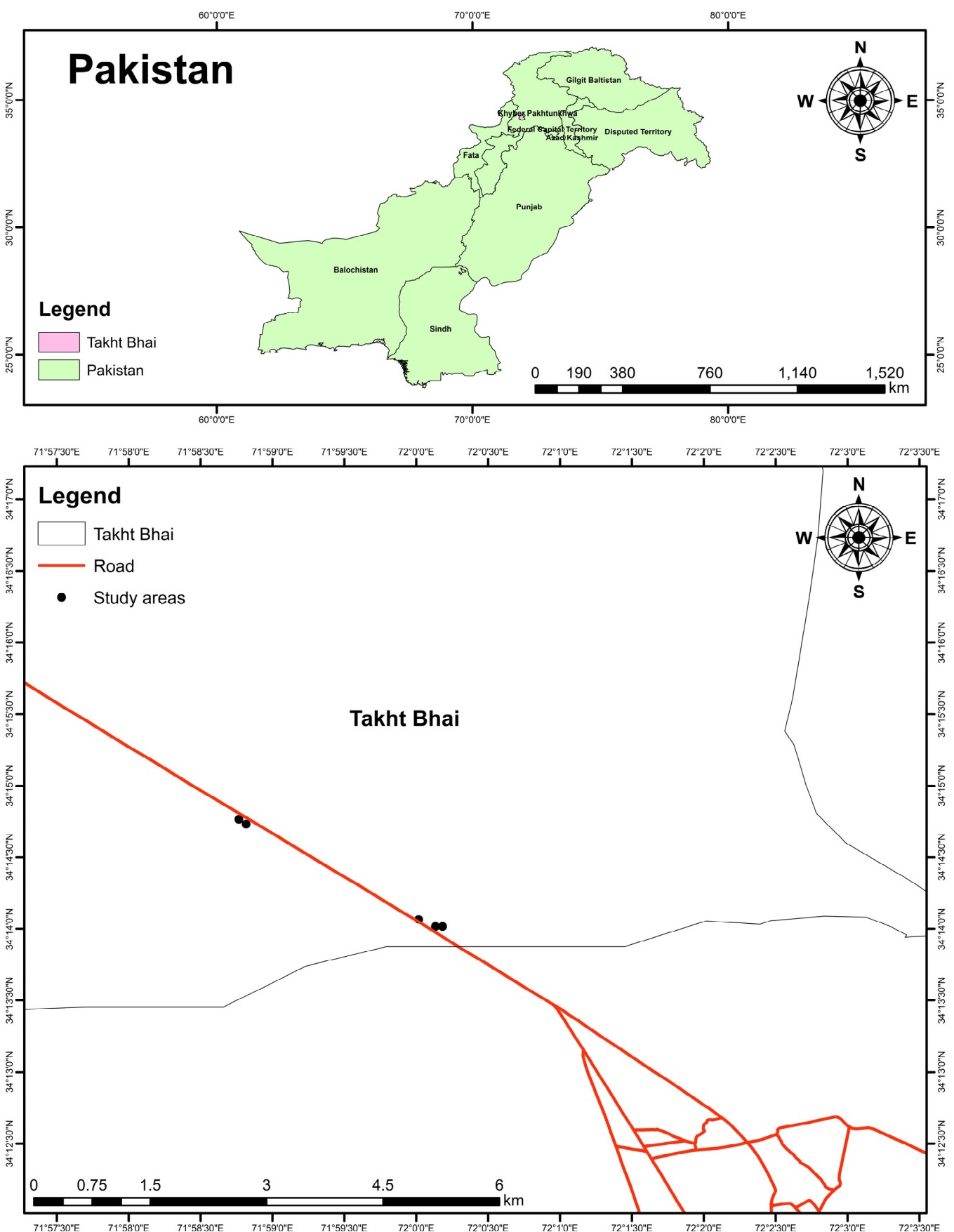

**Figure 1.** Study area map of sampling sites of the plants.

Five different plants species were selected for the current study, i.e., *Fragaria ananassa* (Strawberry), *Lycopersicon esculentum* (Tomato), *Triticum aestivum* (Wheat), *Saccharum officinarum* (Sugar cane), and *Nicotiana tabacum* (Tobacco) (Figure S1). These crops are popularly grown in this area as these are basic and high-income commodities. The plants were cultivated in the growing season under natural field conditions. The selected plants and associated soil samples were collected from each plot. Fifteen replicates for each plant species were collected randomly from each of the selected plots and were transported immediately to the laboratory for further analysis. Plants used per species (3 plots × 15 replicates = 45) and total plants (3 plots × 15 replicates × 5 species = 225). Plants were rinsed with $dH_2O$ to remove the adhered soil and contaminants. Then plants were rinsed with 5 mM Tris HCl pH 6.0 and 5 mM EDTA solution to remove surface-bound metals [36]. All plant samples were arranged into fruits, leaves, stems, and roots and kept in labeled paper bags for further analysis.

### 2.2. Soil and Plant Analysis

The soil samples collected from all sites were kept in the lab. To dry completely. These samples were crushed in pestle and mortar and sieved through 2 mm mesh to remove coarse particles. The fine soil samples obtained were kept in labeled zipper bags for further analysis. The physicochemical parameters, i.e., pH, electrical conductivity (EC), and texture of the samples were evaluated using standard methods of APHA [37]. The dried soil sample was taken in a flask and thoroughly shaken to dissolve in 30 mL of distilled water. Then, the mixture was filtered via Whatman filter paper to collect the filtrate for analyzing EC and pH using an EC meter (Jenco 3173, Chatsworth, CA, USA) and pH meter (Jenco 6175, Chatsworth, CA, USA), respectively. The soil texture was determined by using the hydrometer method. The plant parts were enclosed in labeled envelopes and then kept in an oven at 80 °C for complete drying. Once dried, the samples were blended into powder using a commercial blender and kept in labeled zipper bags in desiccators until further analysis. Powdered samples (0.25 g) of fruit, leaf, stem, and root were taken in conical flasks having a volume of 50 mL. Soil and plant tissues were acid-digested [38]. Briefly, 10 mL $H_2SO_4$ was added into the flasks containing plant or soil sample (0.1 g) and kept overnight in the fume hood. The next day, the flasks were heated on a hot plate set at a high temperature (250 °C) inside the fume hood until black color appeared. Then, $H_2O_2$ was added to the flasks frequently till a clear solution appeared. The digested samples were cooled down at room temperature and the volume was raised to 50 mL using distilled water. It was followed by filtration through Whatman filter paper into labeled bottles. Heavy metal concentrations in all the samples were analyzed using Atomic Absorption Spectrophotometer (SP-IAA320, Jinan, China). Metals translocation and bioconcentration factors were measured to assess the transfer potential of selected HM from soil to the plant [39]. Bio-concentration factor is the ratio of a particular heavy metal in the plant as compared to that in the soil [40]. The translocation factor is the ratio of the concentration of heavy metal in the stem versus the concentration in the root [41].

### 2.3. Physiological and Biochemical Analyses of Plants Samples

For estimation of chlorophyll contents, 200 mg fresh leaves were ground in 2 mL of 80% acetone solution and centrifuged for 5 min at 1000 rpm. The supernatant was pipetted and transferred to a test tube. Further, acetone was added to the test tubes to get a final volume of 6 mL. The obtained extracts were analyzed through a UV-visible spectrophotometer. Aqueous acetone (80%) was run as blank. The absorbance was measured at 663 nm and 645 nm for chlorophyll a and b, respectively [42].

To evaluate carotenoid contents, extraction from leaves was performed with 90% acetone. The absorbance was measured at 480 nm with a spectrophotometer. Acetone (90%) was used as a blank [43]. The carotenoid contents of all the samples were assessed using three biological replicates.

Proline concentrations were assessed according to Bates et al. [44]. Firstly, a 100 mg sample of fresh roots and leaves was weighed in 2 mL tubes and then homogenized with 1.5 mL sulfosalicylic acid (3%). The homogenate was centrifuged at 13,000 rpm for 5 min. Then, a 300 μL aliquot of the clear supernatant was taken and mixed with 2 mL of an equal volume of acetic acid and acid ninhydrin. The mixture was incubated in a boiling water bath for one hour. The reaction was stopped by transferring the tubes from the water bath to the ice bucket. It was followed by adding 1 mL of toluene and vigorous shaking. In the aqueous phase, the colored layer having toluene was taken through a micropipette into a tube, which was kept at room temperature for warming. For proline estimation, the absorbance of the mixture was determined at 520 nm while keeping toluene as a blank/reference. This reaction was run with three replicates for all the samples.

To estimate total phenolics contents, Folin-Ciocalteau (FC) reagent method was used with little modifications [45]. Each sample of 200 mg from overnight air-dried tissues was blended, mixed with 10 mL of ethanol (80%), and stirred vigorously in covered flasks for 30 min. 2 mL aliquot of the extract was centrifuged at 13,000 rpm for 3–5 min. Tenfold diluted FC reagent (250 μL) was added post-centrifugation to 100 μL methanolic extract and kept in dark for 3–5 min at room temperature. It was followed by adding 500 μL of 7% sodium carbonate solution and then DI water was added to get the final volume of 5 mL. Again, the samples were kept in dark for 2 h before measuring the absorbance at 760 nm. 80% methanol was used as a reference solution. Analysis for all samples was performed with three replicates.

### 2.4. Health Risk Assessment

### 2.4.1. Estimated Daily Intake (EDI)

The estimated daily intake of the metals was determined based on their mean concentration in each plant sample and the estimated daily consumption of the vegetables in grams. The EDI value of each metal of interest was determined by the formula used by Chen et al. [46] with slight modification as presented in the following equation:

$$EDI = \frac{E_f \times E_d \times F_{IR} \times C_m \times C_f}{B_w \times T_A} \times 0.001 \tag{1}$$

where $E_f$ is exposure frequency (365 days/year), $E_D$ is the exposure duration (65 years), equivalent to an average lifetime [47], and $F_{IR}$ is the average food (vegetable) consumption (240 g/person/day) which were obtained from the World Health Report (WHO, 2002) for low vegetable intake; $C_M$ is the metal concentration (mg/kg dry weight), $C_f$ is the concentration conversion factor for fresh vegetable weight to dry weight, i.e., 0.085 [48], $B_W$ is reference body weight for an adult, which is 70 kg [47], $T_A$ is the average exposure time (65 years × 365 days) and 0.001 is the unit conversion factor.

### 2.4.2. Target Hazard Quotient (THQ)

The target hazard quotient (THQ) values were estimated to assess non-carcinogenic human health risks from the consumption of vegetables contaminated by heavy metals. The THQ values were calculated using the following equation as described by Chen et al. [46].

$$THQ = \frac{EDI}{RfD} \tag{2}$$

where *EDI* is the estimated daily metal intake of the population in mg/day/kg body weight and *RfD* is the oral reference dose (mg/kg/day) values which were 0.0035 for Pb and 0.001 for Cd (US-EPA). If the value of THQ is <1, it is generally presumed to be safe for the risk of non-carcinogenic effects and if it is >1, it is supposed that there is a chance of non-carcinogenic effects with an increasing probability as the value upsurges [46,49].

### 2.4.3. Hazard Index (HI)

It has been documented that the individual health risks of the analyzed heavy metals in the same vegetable are accumulative and that is expressed as a hazard index [46,49]. Accordingly, the HI of target metals considered in this study was calculated using the following equation proposed by Antoine et al. [49]:

$$HI = \sum_{n=1}^{i} THQ_n; i = 1, 2, 3, \ldots \ldots, n \qquad (3)$$

where *HI* is the sum of various metals hazards. If the *HI* value became <1.0, there is no apparent health impact due to the metals considered. However, an *HI* value of >1.0 indicates potential health impact implications. A serious chronic health impact has been suggested for *HI* > 10.0 [49].

### 2.4.4. Target Cancer Risk (TCR)

The cancer risk posed to human health due to the ingestion of individual possibly carcinogenic metals was estimated using the following equation as described by Sharma et al. [50]. Then, the target cancer risk (TCR) resulting from heavy metals (Pb and Cd) ingestion, which may promote carcinogenic effects depending on the exposure dose, was calculated using the following equation as described by Kamunda et al. [51].

$$CR = EDI \times CPS_o \qquad (4)$$

$$CR = \sum_{n=1}^{i} CR; i = 1, 2, 3, \ldots \ldots, n \qquad (5)$$

where *CR* represents cancer risk over a lifetime by individual heavy metal ingestion, *EDI* is the estimated daily metal intake of the population in mg/day/kg body weight, $CPS_o$ is the oral cancer slope factor in (mg/kg/day)-1 and n is the number of heavy metals considered for cancer risk calculation. The $CPS_o$ values used for Pb and Cd were 0.0035 and 0.001, respectively [51]. It has been pointed out that the slope factor converts the estimated daily intake of the metal averaged over a lifetime of exposure directly to the incremental risk of an individual developing cancer [52].

*BCF* is the ratio of the concentration of a particular heavy metal in the plant to its concentration in the soil [39].

### 2.5. Statistical Analysis

In this study, Microsoft Excel was used to find the means of the replicates and calculate the standard deviations. While ANOVA was done through GraphPad Prism, version 5. Significant difference among different values was obtained through the least significant difference (LSD) test.

## 3. Results and Discussion

### 3.1. Physicochemical Properties of Soil

The soils collected from different spots (plots) based on distance from the main highway and their physicochemical properties are given in Table 1. A slight difference in the pH of soils from the three plots was observed. Sharma and Prasad [53] reported pH values for soil samples from the roadside field almost in the same range. The electrical conductivity was higher (618 μS) in samples collected closest to the road and reduced (591 μS) with increasing distance from the road, which confirms the previous outcomes of the study by Sharma and Prasad [53]. Our current study demonstrates that the concentration of HM in roadside fields has an inverse correlation with distance from the road. Metals concentration (mg kg$^{-1}$) at sites nearest to the road was high and decreased with an increase in distance from the main road. The concentrations of Pb in soil were below 0.3 mg kg$^{-1}$, which is the permissible limit set by the World health organization (WHO), while Cd concentration was higher than the threshold value of 3 mg kg$^{-1}$, which is a standard set by WHO [54] (Table 1).

**Table 1.** Physicochemical properties and heavy metals concentrations in soils sampled at different distances from the main highway.

| Distance from Highway | pH | EC ($\mu$S cm$^{-1}$) | Soil Texture | Lead (mg kg$^{-1}$) | Cadmium (mg kg$^{-1}$) |
|---|---|---|---|---|---|
| 100 m distance | 6.89 ± 0.17 | 591 ± 1.21 | Loamy | 2.67 ± 1.93 | 0.47 ± 0.03 |
| 10–50 m distance | 7.01 ± 0.91 | 612 ± 1.91 | Loamy | 18.84 ± 13.11 | 1.17 ± 0.15 |
| 0–10 m distance | 6.94 ± 0.88 | 618 ± 0.84 | Loamy | 32.02 ± 17.37 | 4.51 ± 0.77 |

### 3.2. Lead (Pb) Concentration in Plant Tissues

The concentration of Pb in various parts of the experimental plants (strawberry, tomato, wheat, tobacco, and sugarcane) is presented in Figure 2A–E. Pb uptake was higher in plants nearest to the main highway. Pb concentration was highest in strawberries compared to other studied plants. The higher Pb concentration in parts observed in order, i.e., roots > leaves > stem > fruits in strawberry, tomato, and tobacco. At the nearest to the road, the strawberry and tomato fruits, leaves of tobacco and stem from sugarcane showed the highest Pb concentrations, which were above the WHO permissible limits (i.e., 0.3 mg kg$^{-1}$). Thus, our results depict that Pb concentration exceeded the threshold level [26] in various parts of the plants including edible fruit sections of strawberries and tomatoes. Khan et al. [55] found a higher concentration of Pb in leaves of certain crop plants such as cauliflower, spinach, tomatoes, and carrot that was above the permissible limit. These results support the findings that plants can accumulate high concentrations of lead in their tissues. Ahsan et al. [56] worked on some edible plants and observed that Pb and Cd contents were above the recommended thresholds in edible portions, while Arora et al. [57] had similar findings in edible portions of vegetables, which is of great concern because of potential health hazards to human beings. In tomato seedlings, Pb uptake, distribution, and accumulation were highest in the root, followed by the leaf, shoot, and fruits. The uptake of HM by crop plants depends upon the type of species as well as the physicochemical characteristics of the soils. Through different mechanisms, Pb accumulation in many plants exceeds a hundred times the maximum threshold levels according to WHO [26], thus posing a threat to public health.

### 3.3. Pb Translocation and Accumulation in Plants

The translocation, accumulation, and bioconcentration of Pb ($\mu$g dry biomass$^{-1}$) in various parts of selected plant species are given in Table 2. It shows that Pb accumulation decreases when the distance from the road increases. The highest accumulation of Pb (i.e., 121.52 $\mu$g dry biomass$^{-1}$) in the whole plant was observed in strawberries at a distance of 0–10 m, followed by sugarcane, tomato, tobacco, and wheat, respectively. Pb accumulation among reference plants (from 100 m distance from the road) was recorded at more than 1 $\mu$g except for wheat and tobacco. In all plants, roots were observed to have the highest Pb accumulation followed by aerial parts. The translocation of Pb from root to above-ground parts of the plant was <1 in all studied plants except sugarcane. The highest translocation of Pb from root to leaf was observed in strawberry, tomato, and tobacco plants, while higher Pb translocation from root to stem was recorded in wheat and sugarcane. The bioconcentration for Pb was also found to be less than one in all the plants. In tomatoes, the highest bioconcentration of Pb (0.70 $\mu$g dry biomass$^{-1}$) was found at 0–10 m from the road. The translocation and bio-concentration of Pb in all plants decreased as the distance from the road increased.

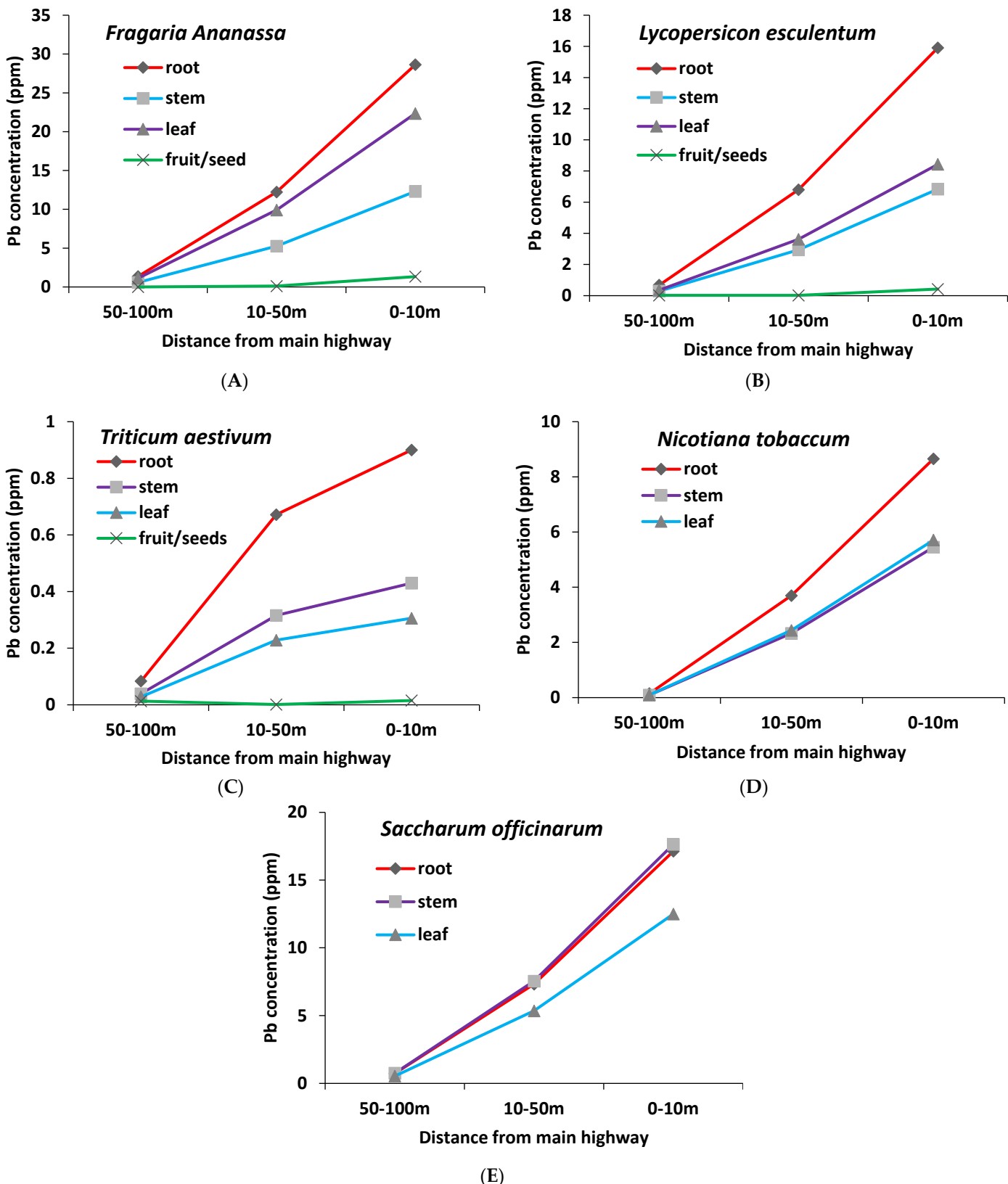

**Figure 2.** (**A–E**) Concentration of Pb (mg kg$^{-1}$) in roots, stems, leaves, and fruits of *Fragaria ananassa* (**A**), *Lycopersicon esculentum* (**B**), *Triticum aestivum* (**C**), *Nicotiana tabacum* (**D**), and *Saccharum officinarum* (**E**) plants. The permissible level for Pb in edible plants is 0.3 mg kg$^{-1}$.

**Table 2.** Pb accumulation (µg dry biomass$^{-1}$), translocation, and bioconcentration in plants (*Fragaria ananassa*, *Lycopersicon esculentum*, *Triticum aestivum*, *Nicotiana tabacum*, and *Saccharum officinarum*. Pb accumulation in roots, stems, leaves, and entire plants (EP). Pb translocation from roots to stems (R–S) and from roots to leaves (R–L).

| Plant | Distance from the Road (m) | Lead (Pb)Accumulation (µg Dry Biomass$^{-1}$) | | | | Pb Translocation | | Pb Bioconcentration |
|---|---|---|---|---|---|---|---|---|
| | | Root | Stem | Leaves | EP | R–S | R–L | |
| *Fragaria ananassa* | 50–100 m | 1.57 ± 0.22 | 1.29 ± 0.18 | 2.45 ± 0.34 | 5.31 ± 0.74 | 0.43 ± 0.05 | 0.83 ± 0.04 | 0.24 ± 0.003 |
| | 10–50 m | 9.53 ± 2.45 | 5.91 ± 1.52 | 16.14 ± 4.15 | 31.58 ± 8.13 | 0.48 ± 0.03 | 0.81 ± 0.07 | 0.30 ± 0.002 |
| | 0–10 m | 50.80 ± 16.9 | 22.22 ± 7.42 | 48.50 ± 16.2 | 121.52 ± 40.5 | 0.38 ± 0.08 | 0.78 ± 0.06 | 0.34 ± 0.010 |
| *Lycopersicon esculentum* | 50–100 m | 0.78 ± 0.11 | 0.65 ± 0.09 | 0.78 ± 0.11 | 2.21 ± 0.31 | 0.45 ± 0.06 | 0.59 ± 0.04 | 0.56 ± 0.011 |
| | 10–50 m | 5.29 ± 1.36 | 3.28 ± 0.84 | 5.87 ± 1.51 | 14.45 ± 3.72 | 0.53 ± 0.03 | 0.53 ± 0.05 | 0.64 ± 0.010 |
| | 0–10 m | 28.22 ± 9.42 | 12.34 ± 4.12 | 18.31 ± 6.11 | 58.87 ± 19.66 | 0.42 ± 0.07 | 0.45 ± 0.05 | 0.70 ± 0.008 |
| *Triticum aestivum* | 50–100 m | 0.10 ± 0.01 | 0.09 ± 0.01 | 0.06 ± 0.01 | 0.25 ± 0.03 | 0.47 ± 0.04 | 0.31 ± 0.05 | 0.11 ± 0.003 |
| | 10–50 m | 0.52 ± 0.13 | 0.35 ± 0.09 | 0.37 ± 0.10 | 1.25 ± 0.32 | 0.41 ± 0.08 | 0.36 ± 0.06 | 0.12 ± 0.005 |
| | 0–10 m | 1.60 ± 0.53 | 0.76 ± 0.26 | 0.66 ± 0.22 | 3.03 ± 1.01 | 0.43 ± 0.06 | 0.39 ± 0.03 | 0.08 ± 0.000 |
| *Nicotiana tabacum* | 50–100 m | 0.17 ± 0.02 | 0.21 ± 0.03 | 0.21 ± 0.03 | 0.59 ± 0.08 | 0.63 ± 0.02 | 0.66 ± 0.08 | 0.12 ± 0.008 |
| | 10–50 m | 2.88 ± 0.74 | 2.62 ± 0.67 | 3.98 ± 1.02 | 9.47 ± 2.44 | 0.71 ± 0.05 | 0.60 ± 0.05 | 0.14 ± 0.003 |
| | 0–10 m | 15.36 ± 5.13 | 9.84 ± 3.29 | 12.41 ± 4.14 | 37.60 ± 12.56 | 0.66 ± 0.02 | 0.71 ± 0.04 | 0.16 ±0.001 |
| *Saccharum officinarum* | 50–100 m | 0.84 ± 0.12 | 1.66 ± 0.23 | 1.16 ± 0.16 | 3.67 ± 0.51 | 1.01 ± 0.08 | 0.73 ± 0.07 | 0.44 ± 0.000 |
| | 10–50 m | 5.70 ± 1.47 | 8.46 ± 2.18 | 8.70 ± 2.24 | 22.86 ± 5.88 | 1.12 ± 0.10 | 0.79 ± 0.09 | 0.34 ± 0.001 |
| | 0–10 m | 30.38 ± 10.1 | 31.83 ± 10.6 | 27.15 ± 9.07 | 89.35 ± 29.84 | 1.03 ± 0.09 | 0.71 ± 0.03 | 0.43 ± 0.000 |

The values following the mean values as ± represent the standard deviation where *n* = 3.

### 3.4. Cadmium (Cd) Concentration in Plant Tissues

Cd concentrations in different parts of the five studied plants are shown in Figure 3A–E. The concentration of Cd in plants reduced with increasing distance from the road, i.e., 0–10 m > 10–50 m > 100 m. In strawberry plants, the concentration of Cd was highly significant. Like Pb, Cd concentration was highest in roots, followed by leaves in strawberries, tomato, and tobacco; however, in the case of wheat and sugarcane, it was higher in stems. Strawberry fruits and tobacco leaves have Cd concentrations above permissible levels [26]. The diminution in Cd concentrations with increasing distance from the road exhibited that emissions from automobiles contribute a substantial amount of metals in the roadside farmlands [58]. Ahsan et al. [56] observed Cd contents in edible parts of plants that were not according to the threshold limits of the WHO. Similarly, Liu et al. [59] reported higher concentrations of toxic HM in vegetables. Therefore, Cd build-up in edible portions of crops and vegetables is of increasing concern.

### 3.5. Cd Translocation and Bioconcentration in Plant Parts

The translocation, accumulation, and bioconcentration (µg dry biomass$^{-1}$) of various parts of the studied plants are shown in Table 3. From the table, it can be illustrated that the highest accumulation of Cd (20.85 µg dry biomass$^{-1}$) was found in strawberries, while the lowest accumulation of Cd was recorded in wheat at the site nearest to the road.

Cd translocation in different parts was found < 1 in all plants except sugarcane. Translocation of Cd from roots to stem at nearest soil to road was higher in sugarcane (i.e., 1.13 µg dry biomass$^{-1}$), while translocation from root to leaf was highest in strawberry, which is 0.78 µg dry biomass$^{-1}$. In sugar cane, the translocation factor- for Cd from root to stem (1.02 µg dry biomass$^{-1}$), and from root to leaves (0.71 µg dry biomass$^{-1}$) was the highest at a distance of 100 m from the road. Cd bioconcentration was found < 1 in selected plants in all the sites except for tobacco at 0-10 m distance, which was 1.08 µg dry biomass$^{-1}$. At all the selected distances from the road, Cd accumulation, translocation as well as bioconcentration decreased with increasing distance from the road. The BCF for Cd was higher compared to Pb in selected crop plants, with the highest value recorded in tobacco and sugarcane followed by strawberry, while the lowest concentrations have been noticed in wheat (Tables 2 and 3).

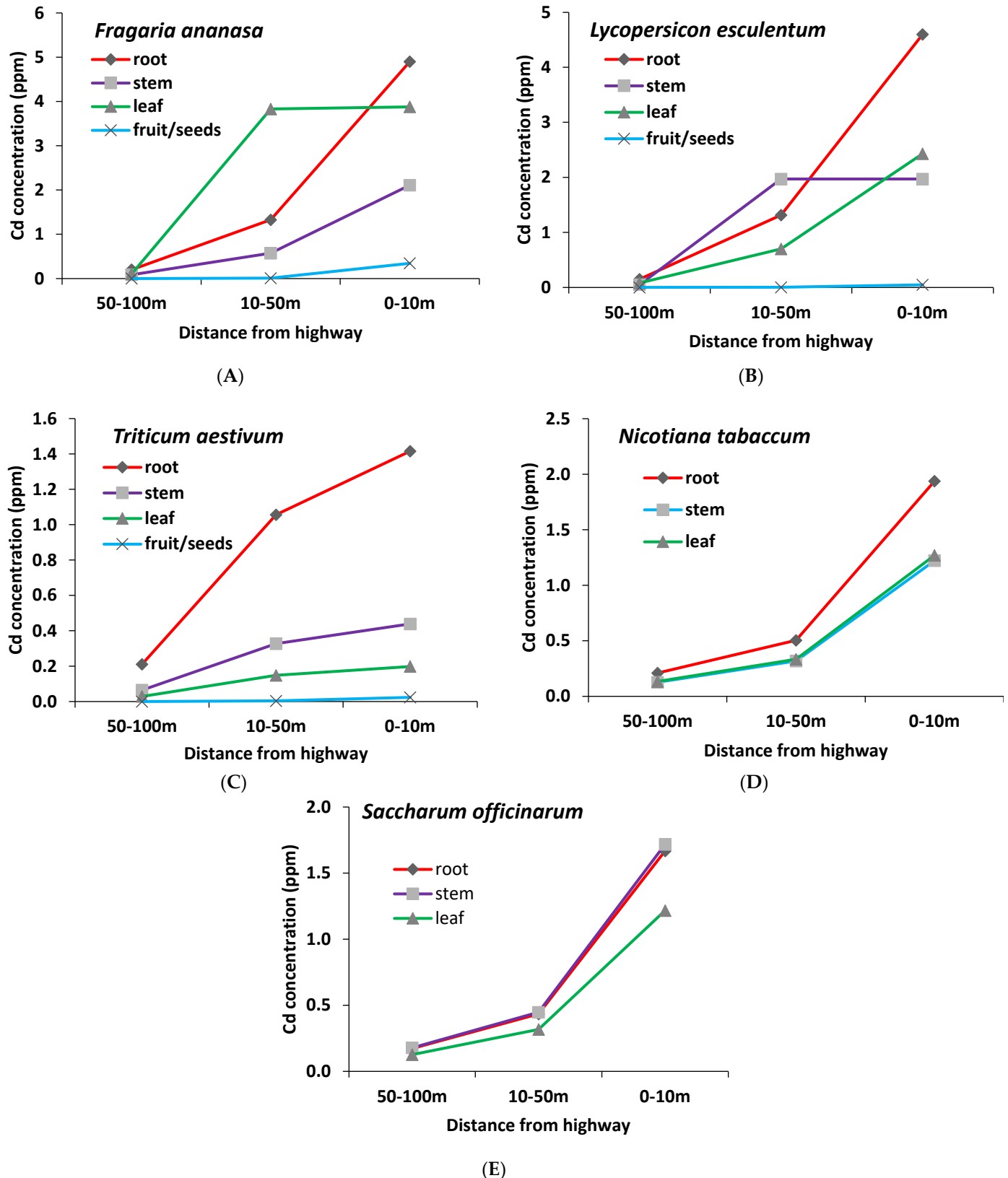

**Figure 3.** (**A–E**) Concentration of Cd (mg kg$^{-1}$) in root, stem, leaves, and fruits of *Fragaria ananassa* (**A**), *Lycopersicon esculentum* (**B**), *Triticum aestivum* (**C**), *Nicotiana tabacum* (**D**), and *Saccharum officinarum* (**E**). The permissible level for Cd in edible plants is 0.1 mg kg$^{-1}$.

**Table 3.** Cd translocation and accumulation ($\mu$g dry biomass$^{-1}$) in plants (*Fragaria ananassa, Lycopersicon esculentum, Triticum aestivum, Nicotiana tabacum,* and *Saccharum officinarum).* Accumulation of Cd in roots, stems, leaves, and entire plants (EP). Translocation of Cd from root to stem (R–S) and from root to leaves (R–L).

| Plant | Distance from Road | Cd Accumulation ($\mu$g Dry Biomass$^{-1}$) | | | | Cd Translocation | | Cd Bioconcentration |
| | | Root | Stem | Leaves | EP | R–S | R–L | |
|---|---|---|---|---|---|---|---|---|
| *Fragaria ananassa* | 50–100 m | 0.23 ± 0.03 | 0.19 ± 0.03 | 0.23 ± 0.03 | 0.66 ± 0.09 | 0.43 ± 0.02 | 0.53 ± 0.05 | 0.80 ± 0.04 |
| | 10–50 m | 1.03 ± 0.27 | 0.64 ± 0.16 | 1.75 ± 0.45 | 3.42 ± 0.88 | 0.55 ± 0.08 | 0.81 ± 0.03 | 0.83 ± 0.05 |
| | 0–10 m | 8.71 ± 2.91 | 8.32 ± 1.27 | 3.81 ± 2.78 | 20.85 ± 6.96 | 0.40 ± 0.04 | 0.78 ± 0.08 | 0.85 ± 0.07 |
| *Lycopersicon esculentum* | 50–100 m | 0.17 ± 0.02 | 0.11 ± 0.02 | 0.17 ± 0.02 | 0.46 ± 0.06 | 0.33 ± 0.03 | 0.51 ± 0.03 | 0.58 ± 0.03 |
| | 10–50 m | 1.02 ± 0.26 | 0.63 ± 0.16 | 1.13 ± 0.29 | 2.79 ± 0.72 | 0.43 ± 0.06 | 0.58 ± 0.05 | 0.65 ± 0.08 |
| | 0–10 m | 8.16 ± 2.72 | 5.29 ± 1.19 | 3.14 ± 1.77 | 17.01 ± 5.68 | 0.45 ± 0.24 | 0.53 ± 0.02 | 0.68 ± 0.05 |
| *Triticum aestivum* | 50–100 m | 0.24 ± 0.03 | 0.14 ± 0.02 | 0.06 ± 0.01 | 0.45 ± 0.06 | 0.31 ± 0.06 | 0.14 ± 0.01 | 0.17 ± 0.07 |
| | 10–50 m | 0.82 ± 0.21 | 0.37 ± 0.09 | 0.24 ± 0.06 | 1.43 ± 0.37 | 0.39 ± 0.03 | 0.19 ± 0.03 | 0.14 ± 0.07 |
| | 0–10 m | 2.51 ± 0.84 | 0.79 ± 0.26 | 0.43 ± 0.14 | 3.73 ± 1.25 | 0.33 ± 0.07 | 0.21 ± 0.02 | 0.35 ± 0.04 |
| *Nicotiana tabacum* | 50–100 m | 0.23 ± 0.03 | 0.28 ± 0.04 | 0.29 ± 0.04 | 0.80 ± 0.11 | 0.65 ± 0.08 | 0.66 ± 0.03 | 0.88 ± 0.08 |
| | 10–50 m | 0.40 ± 0.22 | 0.37 ± 0.20 | 0.56 ± 0.30 | 1.33 ± 0.72 | 0.71 ± 0.05 | 0.69 ± 0.06 | 0.94 ± 0.07 |
| | 0–10 m | 3.44 ± 1.15 | 2.20 ± 0.74 | 2.78 ± 0.93 | 8.42 ± 2.81 | 0.74 ± 0.03 | 0.63 ± 0.09 | 1.08 ± 0.09 |
| *Saccharum officinarum* | 50–100 m | 0.20 ± 0.03 | 0.40 ± 0.06 | 0.27 ± 0.04 | 0.87 ± 0.12 | 1.02 ± 0.09 | 0.71 ± 0.05 | 0.85 ± 0.03 |
| | 10–50 m | 0.34 ± 0.12 | 0.51 ± 0.18 | 0.52 ± 0.18 | 1.37 ± 0.48 | 1.08 ± 0.10 | 0.83 ± 0.07 | 0.95 ± 0.05 |
| | 0–10 m | 3.10 ± 0.99 | 2.96 ± 1.04 | 2.64 ± 0.88 | 8.70 ± 2.91 | 1.13 ± 0.08 | 0.74 ± 0.05 | 0.91 ± 0.08 |

The values following the mean values as ± represent the standard deviation where $n = 3$.

### 3.6. Proline Contents in Plants at Various Distances from the Main Highway

To cope with the stress of the HM, plants accumulate free proline to prevent oxidative stress. It is regarded to have strong antioxidative potential due to which it prevents plant cell death [60]. Our present study demonstrates that both the studied HM triggered proline synthesis in the roots and leaves of plants collected at all three sites at varying distances from the road, as presented in Table 4. The results revealed that proline accumulation was higher in root tissues among all experimental plants compared to their leaves. In the nearest range, a very high concentration of proline was detected; however, its levels started to decrease as the distance from the road increased. A strawberry from site 0–10 m distance exhibited proline accumulation of 3.33 ppm, which is highly significant, whereas wheat shows a lower proline content of 0.90 ppm. In the case of leaves, the pattern of proline contents accumulation was similar, i.e., highest in strawberry (2.33 ppm) and lowest in wheat (0.72 ppm).

**Table 4.** Proline, carotenoids, phenolic, and chlorophyll contents (ppm) in various plants parts.

| Plants | Distance from Road | Proline (ppm) | | Phenolics (ppm) | | Carotenoids (ppm) | Chlorophylls (ppm) | | |
| | | Root | Leaves | Root | Leaves | | A | b | a + b |
|---|---|---|---|---|---|---|---|---|---|
| *Fragaria ananassa* | 50–100 m | 1.33 ± 0.09 [c] | 0.93 ± 0.07 [c] | 0.25 ± 0.02 [c] | 0.45 ± 0.03 [c] | 0.61 ± 0.02 [a] | 11.11 ± 0.78 [a] | 5.22 ± 0.37 [a] | 16.33 ± 1.14 [a] |
| | 10–50 m | 2.00 ± 0.26 [b] | 1.40 ± 0.18 [b] | 0.30 ± 0.04 [b] | 0.60 ± 0.08 [b] | 0.45 ± 0.06 [b] | 10.00 ± 1.30 [b] | 3.65 ± 0.47 [b] | 13.65 ± 1.77 [b] |
| | 0–10 m | 3.33 ± 0.57 [a] | 2.33 ± 0.40 [a] | 0.51 ± 0.09 [a] | 1.34 ± 0.23 [a] | 0.32 ± 0.10 [c] | 8.00 ± 1.36 [c] | 2.61 ± 0.44 [c] | 10.61 ± 1.80 [c] |
| *Lycopersicon esculentum* | 50–100 m | 0.97 ± 0.07 [c] | 0.78 ± 0.05 [c] | 0.07 ± 0.00 [c] | 0.12 ± 0.01 [c] | 0.52 ± 0.03 [a] | 10.23 ± 0.72 [a] | 4.99 ± 0.35 [a] | 15.22 ± 1.07 [a] |
| | 10–50 m | 1.56 ± 0.20 [b] | 1.25 ± 0.16 [b] | 0.08 ± 0.01 [b] | 0.16 ± 0.02 [b] | 0.40 ± 0.05 [b] | 9.20 ± 1.20 [b] | 3.75 ± 0.49 [b] | 12.95 ± 1.68 [b] |
| | 0–10 m | 1.95 ± 0.33 [a] | 1.56 ± 0.27 [a] | 0.14 ± 0.02 [a] | 0.36 ± 0.06 [a] | 0.39 ± 0.09 [c] | 7.36 ± 1.25 [c] | 2.50 ± 0.42 [c] | 9.86 ± 1.68 [c] |
| *Triticum aestivum* | 50–100 m | 0.45 ± 0.03 [c] | 0.36 ± 0.03 [c] | 0.17 ± 0.01 [c] | 0.30 ± 0.02 [c] | 0.48 ± 0.02 [a] | 8.18 ± 0.57 [a] | 4.11 ± 0.29 [a] | 12.29 ± 0.86 [a] |
| | 10–50 m | 0.72 ± 0.09 [b] | 0.58 ± 0.07 [b] | 0.20 ± 0.03 [b] | 0.40 ± 0.05 [b] | 0.40 ± 0.06 [b] | 7.36 ± 0.96 [b] | 3.20 ± 0.42 [b] | 10.57 ± 1.37 [b] |
| | 0–10 m | 0.90 ± 0.15 [a] | 0.72 ± 0.12 [a] | 0.34 ± 0.06 [a] | 0.89 ± 0.15 [a] | 0.33 ± 0.07 [c] | 5.89 ± 1.00 [c] | 2.05 ± 0.35 [c] | 7.94 ± 1.35 [c] |
| *Nicotiana tabacum* | 50–100 m | 1.17 ± 0.08 [c] | 0.93 ± 0.07 [c] | 0.09 ± 0.01 [c] | 0.16 ± 0.01 [c] | 0.48 ± 0.02 [a] | 5.79 ± 0.41 [a] | 3.39 ± 0.24 [a] | 9.18 ± 0.64 [a] |
| | 10–50 m | 1.87 ± 0.24 [b] | 1.49 ± 0.19 [b] | 0.11 ± 0.01 [b] | 0.22 ± 0.03 [b] | 0.39 ± 0.06 [b] | 5.61 ± 0.73 [b] | 3.32 ± 0.43 [b] | 8.93 ± 1.16 [b] |
| | 0–10 m | 2.33 ± 0.40 [a] | 1.87 ± 0.32 [a] | 0.19 ± 0.03 [a] | 0.49 ± 0.08 [a] | 0.25 ± 0.07 [c] | 5.22 ± 0.89 [c] | 2.71 ± 0.46 [c] | 7.93 ± 1.35 [c] |
| *Saccharum officinarum* | 50–100 m | 1.38 ± 0.10 [c] | 1.10 ± 0.08 [c] | 0.16 ± 0.01 [c] | 0.28 ± 0.02 [c] | 0.33 ± 0.01 [a] | 7.34 ± 0.51 [a] | 3.13 ± 0.22 [a] | 10.47 ± 0.73 [a] |
| | 10–50 m | 2.21 ± 0.29 [b] | 1.77 ± 0.23 [b] | 0.19 ± 0.02 [b] | 0.37 ± 0.05 [b] | 0.30 ± 0.04 [b] | 6.61 ± 0.86 [b] | 2.50 ± 0.33 [b] | 9.11 ± 1.18 [b] |
| | 0–10 m | 2.76 ± 0.47 [a] | 2.21 ± 0.38 [a] | 0.31 ± 0.05 [a] | 0.83 ± 0.14 [a] | 0.16 ± 0.05 [c] | 5.28 ± 0.90 [c] | 1.56 ± 0.27 [c] | 6.85 ± 1.16 [c] |

The values following the mean values as ± represent the standard deviation where $n = 3$. Mean values sharing the same letter (s) in a column are statistically non-significant with each other at $p \leq 0.05$

Similar findings have been previously documented for *Triticum aestivum* [61]. Bhattacharjee and Mukherjee [62] reported a higher accumulation of proline contents in roots of *Vigna unguiculata* compared to leaves under Cd and Pb stress. Proline accumulation in response to toxic HM has also been reported in tomatoes [63] and wheat plants [64], and certain weed plants [65]. From the present study, it is evident that Pb and Cd-induced stress in our experimental plants resulted in enhanced proline accumulation in their roots and leaves to survive under metal stress. We also found a strong positive correlation between

proline accumulation in plants with both Pb and Cd concentrations. Proline prevents the inactivation of key enzymes by toxic metal ions [66,67]. In addition, proline acts as an osmoregulator and scavenger of free radicals thus protecting plants from oxidative injuries [68]. In this study, high production of proline confirmed that detoxification of ROS enables the plants to tolerate HM stress.

### 3.7. Phenolic Contents in Plants

The plants' exposure to HM stress may induce the production of a high level of phenolics [69]. Phenolic compounds act as potent antioxidants due to their ability to chelate HM and act as ROS quenchers and membrane stabilizers [70]. In our study, a noticeable rise in the production of phenolic content was observed in leaves compared to roots (Table 4). Phenolic content accumulation was negatively correlated to the distance from the road. At the nearest site, the phenolic contents in strawberry leaves were significantly high, i.e., 1.34 ppm, while tomato leaves showed the lowest accumulation of 0.36 ppm phenolic contents. Similarly, strawberry roots accumulated the highest phenolic contents (0.51 ppm) in an order of strawberry > wheat > sugarcane > tobacco > tomato. Rastgoo and Alemzadeh [71] identified that Gouan (*Aeluropus littoralis*) produced the highest amount of phenolic compounds in response to Pb and Cd, as compared to control plants. *Cannabis sativa* and *Ricinus communis* accumulated increased phenolics under Cd stress [72,73]. The highest accumulation of phenolic compounds was observed in the plant on exposure to HM and these compounds are known for their antioxidant activity. The ability of redox reactions enables them to play significant functions as hydrogen donors, reducing agents, reactive oxygen species (ROS) quenchers, and metal ions chelators [71].

### 3.8. Carotenoid and Chlorophyll Contents in Plants

Heavy metals are known to negatively affect the chlorophyll and carotenoid contents of plants, and adversely affect photosynthesis. Chl a, Chl b, total chlorophyll, and carotenoid contents in selected plants at various distances from the main road were assessed in this study. The current study shows that these contents significantly decreased with increasing metal (Pb and Cd) concentrations in plant tissues as shown in Table 4. The chlorophyll and carotenoid contents in plants exhibited the same trend of under Pb and Cd exposure as demonstrated by Emanuil et al. [74] and Rahman et al. [75]. We observed a decrease in carotenoid contents when the Pb and Cd concentrations were higher. A similar trend has been observed for chlorophyll contents in *Aeluropus littoralis* [71]. Öncel et al. [76] reported that both chlorophyll 'a' and 'b' of two varieties decreased significantly under Pb and Cd treatments. Similar findings have also been reported by Ullah et al. [65], John et al. [77], and Mobin and Khan. [78]. Vijayarengan [43] found that increased zinc level in the soil results in decreasing the chlorophyll and carotenoid contents in the leaves of radish plant. A similar change in chlorophyll contents was recorded with Pb and Cd treatments [74,75]. The carotenoid contents decreased with increasing Pb, Cd concentrations demonstrated in various treatments.

### 3.9. Correlations among Different Parameters

The proline and phenolics content of the leaves and roots showed a significantly positive correlation while photosynthetic pigments (chlorophyll and carotenoids) revealed a negative correlation with Pb and Cd concentration in all experimental plants' tissues (Table 5). The findings of the current research revealed that the study sites have high concentrations of Pb; however, the Cd levels are above the WHO-recommended level in these soils. All the plants considered in our study displayed decreasing trend of metal accumulation in their tissues with an increasing distance from the main road. Pb and Cd levels in edible parts of all the studied plants were above the permissible limits set by WHO. The only exceptions were *Triticum aestivum* (for Pb and Cd) and *Lycopersicon esculentum* (for Cd). Earlier, Mabood et al. [8] recorded a similar trend in different physiological and biochemical parameters in different crop species with heavy metals

**Table 5.** Pearson correlation among various physiological and biochemical parameters in different crop species with heavy metals (Cd and Pb).

| Plant Sample | Heavy Metal | Chlorophyll (Leaf) | Carotenoid (Leaf) | Proline (Root) | Proline (Leaf) | Phenolics (Root) | Phenolics (Leaf) |
|---|---|---|---|---|---|---|---|
| *Fragaria annanasa* | Lead | −0.9627 * | −0.8672 | +0.8679 | +0.9973 | +0.7975 | +0.9802 |
| | Cadmium | −0.9973 | −0.8750 | +0.8881 | +0.9894 | +0.9654 | +0.9224 |
| *Lycopersicon esculentum* | Lead | −0.9776 | −0.8540 | +0.8429 | +0.887 | +0.9674 | +0.956 |
| | Cadmium | −0.9863 | −0.8516 | +0.7293 | +0.9077 | +0.8304 | +0.949 |
| *Triticum aestivum* | Lead | −0.981 | −0.888 | +0.881 | +0.9837 | +0.7326 | +0.8698 |
| | Cadmium | −0.9811 | −0.8789 | +0.7841 | +0.9756 | +0.8809 | +0.889 |
| *Nicotiana tabacum* | Lead | −0.9852 | −0.9688 | +0.7775 | +0.9004 | +0.9281 | +0.9699 |
| | Cadmium | −0.9917 | −0.9824 | +0.6725 | +0.7487 | +0.9045 | +0.9912 |
| *Saccharum officinarum* | Lead | −0.9913 | −0.9767 | +0.7801 | +0.8736 | +0.9548 | +0.9774 |
| | Cadmium | −0.9211 | −0.9192 | +0.6791 | +0.7234 | +0.9456 | +0.9911 |

* All the values either positive or negative were statically significant at $\alpha = 0.05$.

### 3.10. Risk Assessment

#### 3.10.1. Estimated Daily Intake of Metals and Health Risk Assessment

It is crucial to assess human health risks, especially in developing countries such as Pakistan, where mostly wastewater is used for irrigating fields or HM polluted farmlands are used for agricultural activities. Numerous sources have been contributing to the contamination, which ultimately results in metal toxicity in human beings because, in our country, most people consume wheat, fruits, and vegetables in their diet. In the present study, the EDI of Pb and Cd through edible parts of crop plants and their associated health risks have been assessed and presented in Table 6.

**Table 6.** Estimated daily intake (EDI) of metals (mg person$^{-1}$ day$^{-1}$) through edible parts of crop plants and their associated health risks.

| Plant Species | Distance from Road | EDI Pb | EDI Cd | THQ Pb | THQ Cd | HI | CR Pb | CR Cd | TCR |
|---|---|---|---|---|---|---|---|---|---|
| **Strawberry** | 50–100 m | 0.0011 | 0.0001 | 0.311 | 0.122 | 0.434 | $3.815 \times 10^{-6}$ | $1.224 \times 10^{-7}$ | $3.937 \times 10^{-6}$ |
| | 10–50 m | 0.0064 | 0.0007 | **1.824** | 0.697 | **2.521** | $2.235 \times 10^{-5}$ | $6.965 \times 10^{-7}$ | $2.304 \times 10^{-5}$ |
| | 0–10 m | 0.0206 | 0.0035 | **5.889** | **3.535** | **9.424** | $7.213 \times 10^{-5}$ | $3.535 \times 10^{-6}$ | $7.567 \times 10^{-5}$ |
| **Tomato** | 50–100 m | 0.0004 | 0.0001 | 0.119 | 0.082 | 0.201 | $1.459 \times 10^{-6}$ | $8.160 \times 10^{-8}$ | $1.540 \times 10^{-6}$ |
| | 10–50 m | 0.0027 | 0.0005 | 0.762 | 0.513 | **1.275** | $9.333 \times 10^{-6}$ | $5.129 \times 10^{-7}$ | $9.846 \times 10^{-6}$ |
| | 0–10 m | 0.0089 | 0.0025 | **2.552** | **2.457** | **5.009** | $3.126 \times 10^{-5}$ | $2.457 \times 10^{-6}$ | $3.372 \times 10^{-5}$ |
| **Wheat** | 50–100 m | 0.0000 | 0.0001 | 0.012 | 0.058 | 0.071 | $1.530 \times 10^{-7}$ | $5.829 \times 10^{-8}$ | $2.113 \times 10^{-7}$ |
| | 10–50 m | 0.0002 | 0.0002 | 0.060 | 0.178 | 0.238 | $7.344 \times 10^{-7}$ | $1.778 \times 10^{-7}$ | $9.122 \times 10^{-7}$ |
| | 0–10 m | 0.0004 | 0.0004 | 0.118 | 0.356 | 0.474 | $1.448 \times 10^{-6}$ | $3.555 \times 10^{-7}$ | $1.804 \times 10^{-6}$ |
| **Tobacco** | 50–100 m | 0.0001 | 0.0002 | 0.035 | 0.166 | 0.201 | $4 \times 10^{-7}$ | $1.661 \times 10^{-7}$ | $5.945 \times 10^{-7}$ |
| | 10–50 m | 0.0019 | 0.0003 | 0.550 | 0.271 | 0.821 | $6.732 \times 10^{-6}$ | $2.710 \times 10^{-6}$ | $7.003 \times 10^{-6}$ |
| | 0–10 m | 0.0065 | 0.0015 | **1.853** | **1.451** | **3.304** | $2.270 \times 10^{-5}$ | $1.451 \times 10^{-6}$ | $2.415 \times 10^{-5}$ |
| **Sugarcane** | 50–100 m | 0.0008 | 0.0002 | 0.235 | 0.195 | 0.430 | $2.876 \times 10^{-6}$ | $1.953 \times 10^{-7}$ | $3.072 \times 10^{-6}$ |
| | 10–50 m | 0.0050 | 0.0003 | **1.429** | 0.300 | **1.729** | $1.750 \times 10^{-5}$ | $3.002 \times 10^{-7}$ | $1.780 \times 10^{-5}$ |
| | 0–10 m | 0.0172 | 0.0016 | **4.911** | **1.632** | **6.543** | $6.016 \times 10^{-5}$ | $1.632 \times 10^{-6}$ | $6.179 \times 10^{-5}$ |

Where THQ = target hazard quotient, HI = hazard index, CR = cancer risk, and TCR = target cancer risk.

The EDI for Pb and Cd varied from crop to crop. The THQ for Pb and Cd were greater than 1 in all selected crops except wheat plants at all distances. THQ values were higher in crops grown in the fields nearest to the roadsides and the maximum THQ value for Pb and Cd were 5.889 and 3.535, respectively in strawberries. Our results are comparable to previous studies performed in various regions of Pakistan [28,29]. In the case of target cancer risk (TCR), the maximum value was observed in the case of strawberry plants grown at a distance of 0–10 m from the road. Earlier, the values of cancer risk and TCR were decreased in plants with increasing distances [8].

### 3.10.2. Dry Biomass

The dry biomass decreased with a decrease in distance from the highway and the minimum value was found at 0–10 m from the highway (Figure 4A–E). Earlier it has been found that the biomass of the plants decreased under the increasing concentration of heavy metals [74,75]. This premise is supported by the concentration of Pb and Cd in different plant species which were increasing with decreasing distance from the road (Tables 2 and 3). In the plants studied, the level of metal contamination exceeds the safe limit, which presents a serious threat to human health. Thus, it is recommended that such contaminated fields should not be used for agricultural purposes or at least for the production of edible crops. Moreover, the concerned authorities are advised to monitor the concerned areas regularly by testing soil and crops for metal contamination. However, public awareness is also an important initiative to be established. These measures will not only ensure food safety and security but also ascertains good public health. Therefore, the study establishes to assess metal contamination levels and associated risks to human health both locally and globally.

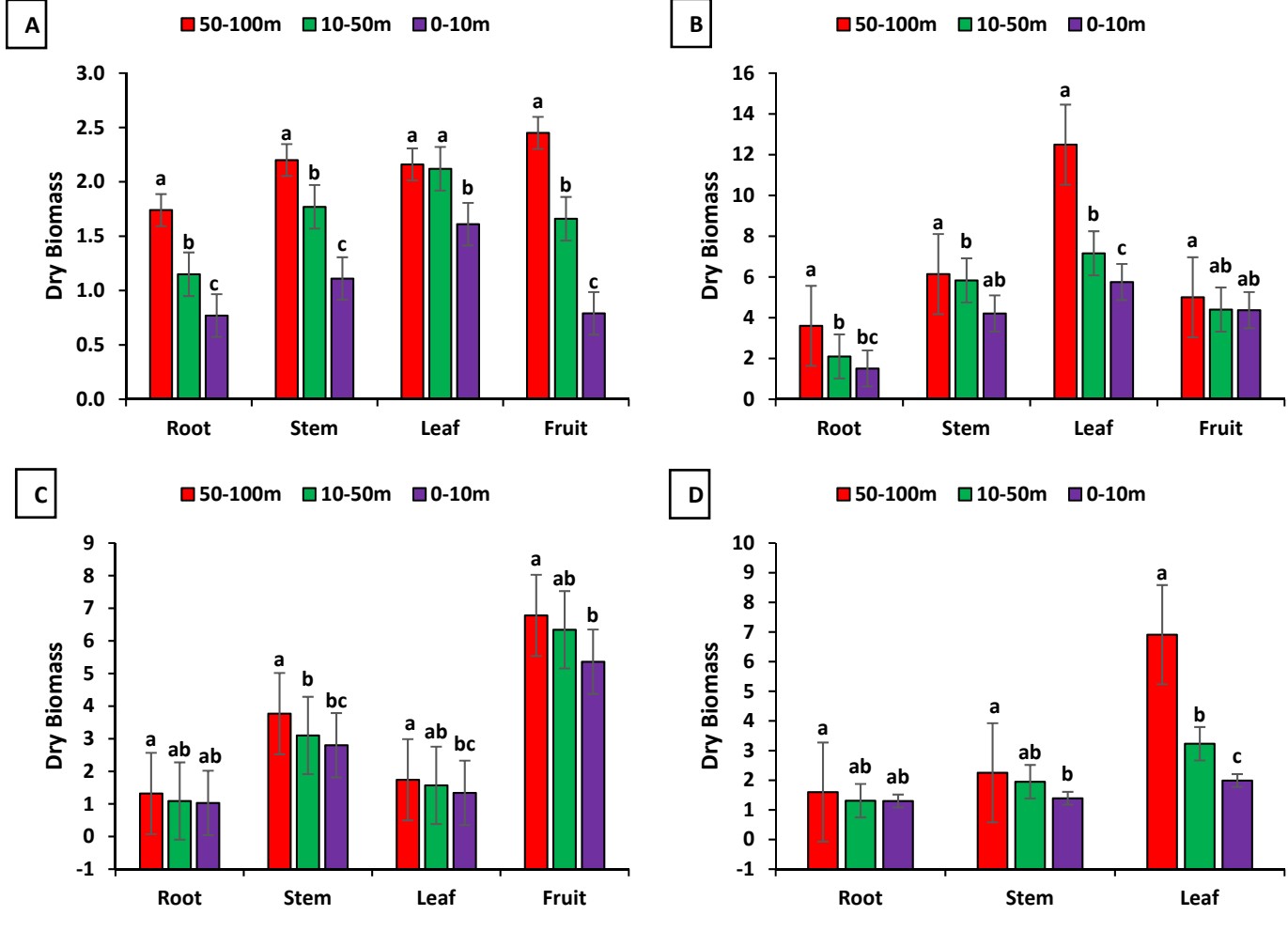

**Figure 4.** *Cont*.

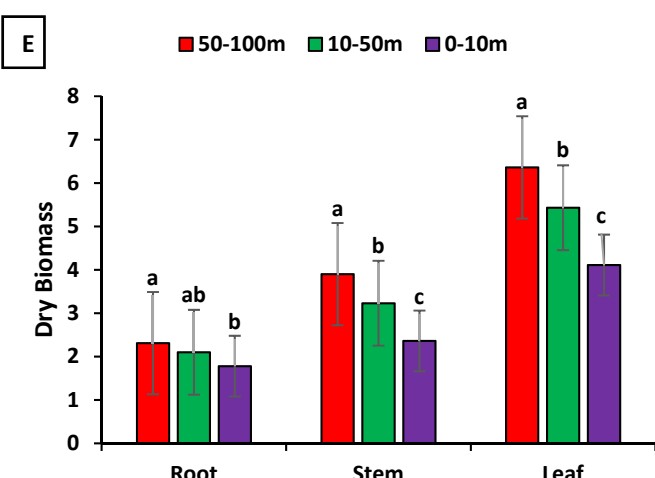

**Figure 4.** (**A–E**) Comparison of dry biomasses of selected plant species (**A**) = Strawberries, (**B**) = Tomatoes, (**C**) = Wheat, (**D**) = Tobacco, and (**E**) = Sugarcane at different distances from the road. The bars with different letters are significantly different from each other at α = 0.05.

## 4. Conclusions

The present study showed that the metal concentrations tend to decrease in all selected plants when the distance from the road increased, i.e., the plants from the site nearest to the road had higher Pb and Cd concentrations and vice versa. Comparatively, Pb concentrations were higher in these plants than in Cd. In our experimental edible crops, Pb and Cd levels were exceeding the threshold values for these metals in edible portions while wheat grains and tomato fruit were exceptions for Pb + Cd and Cd, respectively. Moreover, proline and phenolic contents accumulation in roots and leaves were higher when the metal concentration was higher. The photosynthetic pigments (chlorophyll and carotenoids) in all the plants decreased, as the sampling site was getting closer to the road. The health risk indices such as target hazard quotient (THQ) and hazard index (HI) for edible parts of most plants (strawberry, tomato, tobacco, and sugarcane) are greater than the maximum permissible limits, i.e., >1. Future studies should be conducted to evaluate more toxic HM and other emerging contaminants in agricultural soils and edible crops for risk characterization and assessment. These will be crucial in understanding the metal contamination level and devising certain risk reduction and mitigation strategies.

**Supplementary Materials:** The following supporting information can be downloaded at: https://www.mdpi.com/article/10.3390/su142316263/s1, Figure S1: (A) *Fragaria ananassa* (B) *Lycopersicon esculentum* (C) *Nicotiana tabacum* (D) *Triticum aestivum* and (E) *Saccharum officinarum* plants.

**Author Contributions:** Conceptualization, S.A., F.H. and A.D.; Data curation, S.A., A.U.J. and R.U.; Formal analysis, S.A., F.H., A.U.J., R.U., B.F.A.A. and A.D.; Funding acquisition, B.F.A.A. and A.D.; Investigation, S.A.; Methodology, F.H.; Project administration, F.H., B.F.A.A. and A.D.; Resources, S.A., F.H., A.U.J., R.U., B.F.A.A. and A.D.; Software, F.H., R.U., B.F.A.A. and A.D.; Supervision, F.H. and A.U.J.; Validation, S.A., F.H., A.U.J., R.U., B.F.A.A. and A.D.; Visualization, S.A., F.H., A.U.J., R.U., B.F.A.A. and A.D.; Writing—original draft, S.A.; and Writing—review and editing, F.H., A.U.J., R.U., B.F.A.A. and A.D. All authors have read and agreed to the published version of the manuscript.

**Funding:** The study was financially supported by the Higher Education Commission (HEC) of Pakistan.

**Institutional Review Board Statement:** Not applicable.

**Informed Consent Statement:** Not applicable.

**Data Availability Statement:** Not applicable.

**Acknowledgments:** The Central Resources Laboratory (CRL) at the University of Peshawar and its staff are acknowledged for their support and the facilities provided.

**Conflicts of Interest:** The authors declare no conflict of interest.

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
