# Peer review of "Appraisal of Heavy Metals Accumulation, Physiological Response, and Human Health Risks of Five Crop Species Grown at Various Distances from Traffic Highway"

_sustainability, doi:10.3390/su142316263_

Round 1

Reviewer 1 Report

Dear Editor:

Thank you for giving me the opportunity to revise the MS entitled “Appraisal of toxic metals accumulation, physiological response, and human health risk of five crops species grown at various distances from heavy traffic highway” by Shakeel Ahmad and his/her colleagues that was submitted to “sustainability”. The MS submitted is suitable for sustainability, and some interesting results were showed. However, there are several requirements that have to consider by the authors. In this regard, the following comments are requested to be addressed by the authors:

 The English of the paper is readable; however, I would suggest the authors to have it checked preferably by a native English-speaking person to avoid any mistakes.

Keywords:

Do not use abbreviations for keywords.

 Introduction

 The necessity & novelty of the manuscript should be presented and stressed in the “Introduction” section.

 Materials and Methods

Line 91 Figure 1 is not a geographical map, it is inconsistent with the description of this sentence.

Line 122 The model and other information of the instrument must be provided in detail. Please check the full text carefully

Figure 1 is unclear and unnecessary, please consider deleting or placing it in the SI.

Materials and Methods section should be rewritten, some parts can be merged.

 Results and discussion

 What do those small letters in the table mean? Be clear in the text.

The authors should deepen the discussion.

 Reference

The format of references is not uniform (Line 487Line 489Line510Line540Line550Line 596Line 618).

Some references are not appropriate.

 Line 630 Brassica junceashould be italicized.

  I would suggest that the authors review and include the following studies to improve the manuscript.

1.    He, L.; Su, R.; Chen, Y.; Zeng, P.; Du, L.; Cai, B.; Zhang, A.; Zhu, H., Integration of manganese accumulation, subcellular distribution, chemical forms, and physiological responses to understand manganese tolerance in Macleaya cordata. Environ Sci Pollut R 2022, 29, (26), 39017-39026.

2.    Su, R.; Ou, Q.; Wang, H.; Luo, Y.; Dai, X.; Wang, Y.; Chen, Y.; Shi, L., Comparison of phytoremediation potential of Nerium indicum with inorganic modifier calcium carbonate and organic modifier mushroom residue to lead-zinc tailings. Int. J. Environ. Res. Public Health 2022, 19, (16), 10353.

3.    Han, L.; Chen, Y.; Chen, M.; Wu, Y.; Su, R.; Du, L.; Liu, Z., Mushroom residue modification enhances phytoremediation potential of Paulownia fortunei to lead-zinc slag. Chemosphere 2020, 253, 126774.

Best regards,

Author Response

Response to Reviewer 1 comments

Dear Editor:

Thank you for giving me the opportunity to revise the MS entitled “Appraisal of toxic metals accumulation, physiological response, and human health risk of five crops species grown at various distances from heavy traffic highway” by Shakeel Ahmad and his/her colleagues that was submitted to “sustainability”. The MS submitted is suitable for sustainability, and some interesting results were showed. However, there are several requirements that have to consider by the authors. In this regard, the following comments are requested to be addressed by the authors:

Comment: The English of the paper is readable; however, I would suggest the authors to have it checked preferably by a native English-speaking person to avoid any mistakes.

Response: We have got it proofread by a colleague at UWA regarding any typos and grammatical mistakes

Comment: Keywords: Do not use abbreviations for keywords.

Response: Revised as per suggestion

Introduction

Comment: The necessity & novelty of the manuscript should be presented and stressed in the “Introduction” section.

Response: We have highlighted the novelty of the manuscript as per the suggestion of the reviewer

Materials and Methods

Comment: Line 91 Figure 1 is not a geographical map, it is inconsistent with the description of this sentence.

Response: Please accept our apologies for this mistake. We have added Figure 1 as a study area map and added the plants' pictures as supplementary material as per the suggestion of the worthy reviewer

Comment: Line 122 The model and other information of the instrument must be provided in detail. Please check the full text carefully

Response: Revised accordingly

Comment: Figure 1 is unclear and unnecessary, please consider deleting or placing it in the SI.

Response: We have added the plants' pictures as supplementary material as per the suggestion of the worthy reviewer

Comment: Materials and Methods section should be rewritten, some parts can be merged.

Response: We have revised and merged some parts of the Materials and Methods section as per the suggestion of the reviewer

Results and discussion

Comment: What do those small letters in the table mean? Be clear in the text.

Response: The small letter (m) in the table shows the distance of plant species from roadside

Comment: The authors should deepen the discussion.

Response: Done accordingly

Reference

Comment: The format of references is not uniform (Line 487、Line 489、Line510、Line540、Line550、Line 596、Line 618). Some references are not appropriate.

Response: Revised accordingly

Comment: Line 630 “Brassica juncea” should be italicized.

Response: Revised accordingly

Comment: I would suggest that the authors review and include the following studies to improve the manuscript.

  1. He, L.; Su, R.; Chen, Y.; Zeng, P.; Du, L.; Cai, B.; Zhang, A.; Zhu, H., Integration of manganese accumulation, subcellular distribution, chemical forms, and physiological responses to understand manganese tolerance in Macleaya cordata. Environ Sci Pollut R 2022, 29, (26), 39017-39026.
  2. Su, R.; Ou, Q.; Wang, H.; Luo, Y.; Dai, X.; Wang, Y.; Chen, Y.; Shi, L., Comparison of phytoremediation potential of Nerium indicum with inorganic modifier calcium carbonate and organic modifier mushroom residue to lead-zinc tailings. Int. J. Environ. Res. Public Health 2022, 19, (16), 10353.
  3. Han, L.; Chen, Y.; Chen, M.; Wu, Y.; Su, R.; Du, L.; Liu, Z., Mushroom residue modification enhances phytoremediation potential of Paulownia fortunei to lead-zinc slag. Chemosphere 2020, 253, 126774.

Response: We have consulted the mentioned papers and cited them accordingly

Reviewer 2 Report

Lines 38-39. The word “intensification” is not adequate, it should be replaced or the whole sentence should be rearranged.  

Line 43. The word “synergistically” not necessary not even right, because I don’t think that there is a synergic effect between road surfaces and vehicular traffic in emitting pollutants.

Lines 105-106. “All plant samples were sorted into fruits, leaves, stems, and roots and kept in labelled paper bags for further analysis”. This sentence is no clear. Plant samples were stored into fruits… How can plant samples be stored into fruits? Maybe different plant organs (such as stems, leaves …) were stored?

Line 126. What does it mean the word “Shoot”? It does not seem that it is the right word to be used here.

Line 134. The complete name for the abbreviations EC and WHC should be written when they first appear in the text.

Line 151. The word “reaction” does not seem correctly used. The extraction is not a reaction, if some reaction was involved please explain.

Line 196. The Subtitle should be in Italic. 

Line 339. Write the full name for “ROS” when it first appears in the text.

Table 5.  It is not clear from the table which number represents which correlation. For example the first value 0.9627, does it represent the correlation between Pb and chlorophyll? And the second value -0.8672  Pb-carotenoid, the third +0.8679 Pb-Proline … and so on? Which value represents the correlation “The proline and phenolics content of the leaves and roots showed a significantly positive correlation” (line 367). The table needs to be clearer.

Whyt type of correlation was performed, pearson or other? What is the “p” value for those correlation values?

Figure 4(A-E). The bars are named with letters a, b, c,   why there are sometimes  bars with two letters or with repeated letters?

The name of the study location and country should be mentioned also in abstract and somewhere by the end of introduction part.

Author Response

Response to the Reviewer 2 comments

Comment: Lines 38-39. The word “intensification” is not adequate, it should be replaced or the whole sentence should be rearranged. 

Response: Revised accordingly

Comment: Line 43. The word “synergistically” not necessary not even right, because I don’t think that there is a synergic effect between road surfaces and vehicular traffic in emitting pollutants.

Response: Revised accordingly

Comment: Lines 105-106. “All plant samples were sorted into fruits, leaves, stems, and roots and kept in labelled paper bags for further analysis”. This sentence is no clear. Plant samples were stored into fruits… How can plant samples be stored into fruits? Maybe different plant organs (such as stems, leaves …) were stored?

Response: We have revised the sentence as “Plant samples were arranged/classified into fruits, leaves, stems, and roots and kept in labeled paper bags for further analysis”

Comment: Line 126. What does it mean the word “Shoot”? It does not seem that it is the right word to be used here.

Response: Shoot is replaced with “stem” in the revised manuscript

Comment: Line 134. The complete name for the abbreviations EC and WHC should be written when they first appear in the text.

Response: Revised accordingly

Comment: Line 151. The word “reaction” does not seem correctly used. The extraction is not a reaction, if some reaction was involved please explain.

Response: The sentence has been revised accordingly

Comment: Line 196. The Subtitle should be in Italic.

Response: Revised

Comment: Line 339. Write the full name for “ROS” when it first appears in the text.

Response: Revised accordingly

Comment: Table 5.  It is not clear from the table which number represents which correlation. For example the first value 0.9627, does it represent the correlation between Pb and chlorophyll? And the second value -0.8672  Pb-carotenoid, the third +0.8679 Pb-Proline … and so on? Which value represents the correlation “The proline and phenolics content of the leaves and roots showed a significantly positive correlation” (line 367). The table needs to be clearer. Whyt type of correlation was performed, pearson or other? What is the “p” value for those correlation values?

Response: We have revised the Table 5 description by adding horizontal lines to differentiate the correlation between heavy metals (Cd and Pb) and physiological and biochemical parameters of different plant species among each other. We have also written the statistical significance of each correction in the footnotes of Table 5

Comment: Figure 4(A-E). The bars are named with letters a, b, c, why there are sometimes bars with two letters or with repeated letters?

Response: The bars sharing similar letters show that the mean values are non-significant among each other while different letters show significant differences among the means shown in the form of bars.

Comment: The name of the study location and country should be mentioned also in abstract and somewhere by the end of introduction part.

Response: The name of the study location and country has been added in the material and methods section. We have also mentioned it in the abstract as per the suggestion of the reviewer.

Reviewer 3 Report

The reviewed manuscript is dealing with the appraisal of toxic metals accumulation, physiological response, and human health risk of five crop species grown at various distances from heavy traffic highway. Soil and crop samples were collected at different distances from the main road. Phenolics, carotenoids, chlorophyll, and proline contents in plant parts were assessed. Human health risks were assessed using the USEPA noncarcinogenic risk models. As there is apparently little published about this area, the data have some value. However, the manuscript should be improved. After carefully analyzing the content, I address some major comments, which should be considered prior to acceptance of this manuscript.

1. The authors used different terms (toxic metals, heavy metals) in their study. Please unify the used terms.

2. Figure 1; It will get better to move the provided photo to the supplementary materials and replace it with a location map showing the sampling sites. Consider adding a base map indicating the exact location of the investigated area (Mardan, Pakistan).

3. Materials and Methods; The authors should descript the methods of plant and soil digestion. Also, describe how you measured soil texture.

4. Line 134; WHC, the authors did not use Water Holding Capacity data in their results.

5. Table 1; please consider adding metal concentrations to the title of this table.

6. Lines 215:218; Please revise the used metal permissible limits in soil.

7. Table 6; The calculated DIM and HRI are not logical compared to Pb and Cd concentrations and need to be completely revised. In addition, it will get better to calculate the non-carcinogenic Hazard Question (HI) Hazard Index (HI) and Carcinogenic Risk (CR) considering that Pb and Cd are carcinogens.

8. Accordingly, the abstract and conclusion need to be revised.

9. References style should be revised following the journal instructions.

Author Response

Response to the Reviewer 3 comments

Comment: The reviewed manuscript is dealing with the appraisal of toxic metals accumulation, physiological response, and human health risk of five crop species grown at various distances from heavy traffic highway. Soil and crop samples were collected at different distances from the main road. Phenolics, carotenoids, chlorophyll, and proline contents in plant parts were assessed. Human health risks were assessed using the USEPA noncarcinogenic risk models. As there is apparently little published about this area, the data have some value. However, the manuscript should be improved. After carefully analyzing the content, I address some major comments, which should be considered prior to acceptance of this manuscript.

Response: Thanks for your comments and suggestions. We have revised the whole manuscript as per the suggestions of the worthy reviewers and guest editor

Comment 1. The authors used different terms (toxic metals, heavy metals) in their study. Please unify the used terms.

Response: Revised accordingly

Comment 2. Figure 1; It will get better to move the provided photo to the supplementary materials and replace it with a location map showing the sampling sites. Consider adding a base map indicating the exact location of the investigated area (Mardan, Pakistan).

Response: We have added the study area map. We apologize for any inconvenience caused

Comment 3. Materials and Methods; The authors should descript the methods of plant and soil digestion. Also, describe how you measured soil texture.

Response: We have revised the methods as per the suggestion of the reviewer  

Comment 4. Line 134; WHC, the authors did not use Water Holding Capacity data in their results.

Response: It was a typo and we have removed it in the revised manuscript

Comment 5. Table 1; please consider adding metal concentrations to the title of this table.

Response: We have revised the title to “Physicochemical properties and heavy metals concentrations in soils sampled at different distances from the main highway”

Comment 6. Lines 215:218; Please revise the used metal permissible limits in soil.

Response: Revised

Comment 7. Table 6; The calculated DIM and HRI are not logical compared to Pb and Cd concentrations and need to be completely revised. In addition, it will get better to calculate the non-carcinogenic Hazard Question (THQ) Hazard Index (HI) and Carcinogenic Risk (CR) considering that Pb and Cd are carcinogens.

Response: We have revised the health risk assessment as per the suggestion of the worthy reviewer

Comment 8. Accordingly, the abstract and conclusion need to be revised.

Response: We have the abstract and conclusion accordingly  

Comment 9. References style should be revised following the journal instructions.

Response: Revised accordingly

Round 2

Reviewer 1 Report

ok to accept.

Author Response

Thanks for your recommendation

Reviewer 3 Report

The improvements achieved by the authors in the revision look good. The reviewer appreciates the specific response from the authors, which addressed the reviewer's major concerns. There are some additional minor comments.

1. Figure 1; remove (all other values).

2. Line 236; Please revise the used Pb permissible limits in soil.

3. The calculated health risk must be reflected in the conclusion.

Author Response

Response to Reviewer 3 comments

The improvements achieved by the authors in the revision look good. The reviewer appreciates the specific response from the authors, which addressed the reviewer's major concerns. There are some additional minor comments.

Response: Thanks for your appreciation. We have incorporated all the suggested changes in the revised version

  1. Figure 1; remove (all other values).

Response: Removed

  1. Line 236; Please revise the used Pb permissible limits in soil.

Response: Revised

  1. The calculated health risk must be reflected in the conclusion.

Response: The calculated health risk has been mentioned in the conclusion as per the suggestion of the reviewer